# Perceptual processing in the ventral visual stream requires area TE but not rhinal cortex

Mark AG Eldridge[1]*, Narihisa Matsumoto[2], John H Wittig Jnr[3], Evan C Masseau[1], Richard C Saunders[1], Barry J Richmond[1]*

[1]Laboratory of Neuropsychology, National Institute of Mental Health, National Institutes of Health, Bethesda, United States; [2]Human Informatics Research Institute, National Institute of Advanced Industrial Science and Technology, Tsukuba, Japan; [3]Surgical Neurology Branch, National Institute of Neurological Disorders and Stroke, National Institutes of Health, Bethesda, United States

**Abstract** There is an on-going debate over whether area TE, or the anatomically adjacent rhinal cortex, is the final stage of visual object processing. Both regions have been implicated in visual perception, but their involvement in non-perceptual functions, such as short-term memory, hinders clear-cut interpretation. Here, using a two-interval forced choice task without a short-term memory demand, we find that after bilateral removal of area TE, monkeys trained to categorize images based on perceptual similarity (morphs between dogs and cats), are, on the initial viewing, badly impaired when given a new set of images. They improve markedly with a small amount of practice but nonetheless remain moderately impaired indefinitely. The monkeys with bilateral removal of rhinal cortex are, under all conditions, indistinguishable from unoperated controls. We conclude that the final stage of the integration of visual perceptual information into object percepts in the ventral visual stream occurs in area TE.

DOI: https://doi.org/10.7554/eLife.36310.001

*For correspondence:
mark.a.g.eldridge@gmail.com
(MAGE);
barry.richmond@nih.gov (BJR)

**Competing interests:** The authors declare that no competing interests exist.

## Introduction

An intriguing property of the visual system is how easily and effortlessly we perceive objects (sensory processing), and discriminate among those of similar appearance, even when the particular exemplar has never been seen before. We quickly distinguish any red tomato from any red apple, regardless of the variety. A half-century of behavioral, anatomical and physiological research has revealed that this feat of visual perception is supported by a sequence of connected brain regions stretching from the occipital cortex to inferior temporal cortex (*Figure 1A*) (*Gross et al., 1972*; *Ungerleider and Mishkin, 1982*). Simple features, such as oriented edges or lines, are represented in caudal regions, beginning with area V1 (*Hubel and Wiesel, 1959*). Conjunctions of features defining whole objects are represented in rostral regions, culminating with area TE (*Kobatake and Tanaka, 1994*). Directly adjacent to area TE is rhinal cortex, an anatomically distinct region that receives dense projections from area TE (*Suzuki and Amaral, 1994*), and is thought to be important primarily for memory function (*Meunier et al., 1993*; *Higuchi and Miyashita, 1996*). It has more recently been suggested that rhinal cortex is important for visual perception of complex objects or objects with over-lapping features, and hence might be considered the final stage of the visual perceptual processing hierarchy (*Buckley et al., 2001*; *Bussey et al., 2003*; *Baxter, 2009*), but see *Hampton, 2005*, *Suzuki (2009)*.

To determine whether visual object processing is finalized in area TE, or whether it extends to rhinal cortex, categorization was tested at several levels of perceptual difficulty in a series of experiments using stimuli with overlapping features. All tasks required remembering visual perceptual

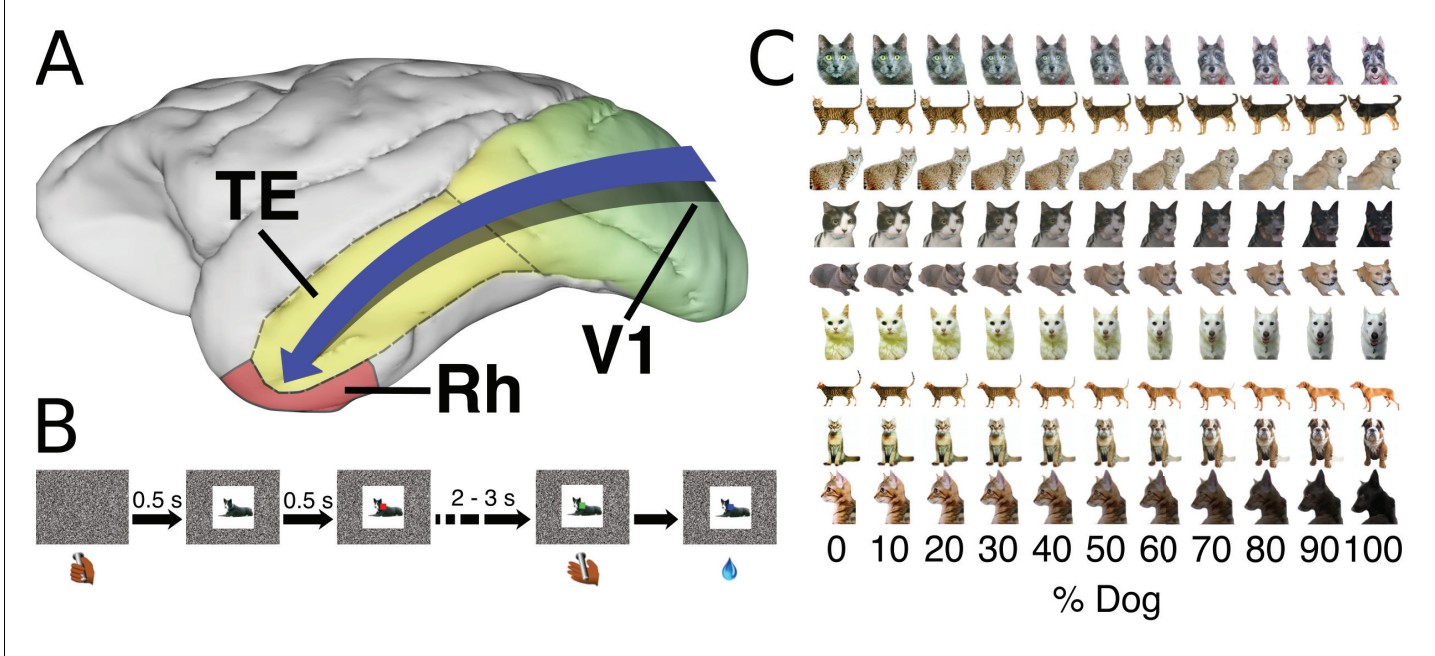

**Figure 1.** Background and task. (**A**) Ventral visual stream - simple features represented in primary visual cortex (green). Increasing complexity of representations in intermediate areas, culminating in the representation of whole objects in area TE (yellow, bounded by dashed line). Immediately rostro-ventral to TE is rhinal cortex (Rh) (red, bounded by solid line - n.b. medial portion of rhinal cortex not visible from this angle). See figure supplement for rhinal cortex reconstructions. (**B**) A single trial from the perceptual categorization task (see supplemental methods for details). (**C**) Examples of the cat-dog morphed images presented as visual stimuli in Experiment 1.
DOI: https://doi.org/10.7554/eLife.36310.002

The following figure supplement is available for figure 1:

**Figure supplement 1** . Estimates of the extent of the aspiration lesions of the three monkeys in the TE-lesioned group (top), and rhinal-lesioned group (bottom) are plotted on coronal sections at the indicated levels, and reconstructed onto lateral/ventral views of the macaque brain, respectively; reconstructions for each case are shown at the bottom of each column.
DOI: https://doi.org/10.7554/eLife.36310.003

categories. However, in every trial, the monkeys responded while the stimulus was present, thereby minimizing demands on short-term memory.

## Results

We tested three groups of three monkeys: an unoperated control group, a group with a bilateral removal of area TE, and a group with a bilateral removal of rhinal cortex (including peri- and ento-rhinal cortex). Monkeys were trained to perform a cat vs. dog category discrimination in a visually cued two-interval forced choice (2-IFC) paradigm (*Figure 1B*) (*Matsumoto et al., 2016*). They were required to judge whether an image was more dog-like or cat-like when presented with stimuli drawn from a set of category-ambiguous 'morphed' images, created by blending and warping cats with dogs in different proportions. The monkeys responded while a stimulus was present, thereby minimizing demands on short-term memory.

### Experiment 1 – learning to categorize morphed stimuli

In Experiment 1, monkeys were presented with a set of stimuli comprising cats, dogs, and intermediate morphed images spaced at 10% increments on an arbitrary scale (Abrosoft, Beijing, China) from 0 % to 100% dog (*Figure 1C*). Monkeys could avoid an extended inter-trial delay by releasing the bar in the first interval (signaled by a red target) for stimuli that were less than 50% dog, and were rewarded for releasing the bar in the second interval (signaled by a green target) for stimuli that were more than 50% dog. This amounts to an asymmetrical reward structure. They were rewarded randomly for releasing during the green interval for 50 – 50 morphs. The monkeys in all three groups

classified most stimuli well the first time they were presented (*Figure 2A,B,C*). The control and rhinal-removal groups improved quickly with repetition, reaching asymptotic performance by the 10th repeat of the stimulus set (*Figure 2D*). The TE-removal group was slower to learn, only reaching the level of performance of controls by the 14th repetition of the stimulus set. Across presentations, the group with TE removals categorized less accurately than the control group (Linear Mixed Effects model, LME, p=0.0072, $z = 2.69$). The deficit seen in the TE-removal group might be attributable to slower learning, as there was an interaction effect between number of presentations by treatment group (Ctl vs. TE) by morph level (LME, p=$1.07 \times 10^{-8}$, $z = -5.72$). The performance of the group with rhinal removals was indistinguishable from that of controls (LME, p=0.23, $z = -1.19$).

## Experiment 2 – area TE removal impairs perceptually difficult categorization

In Experiment 2, we examined sensitivity to perceptual ambiguity at the category boundary, where classification should be most difficult. The stimuli used in Experiment 2 were derived from the same cat and dog pairs used in Experiment 1. We continued to explore the full range of category space

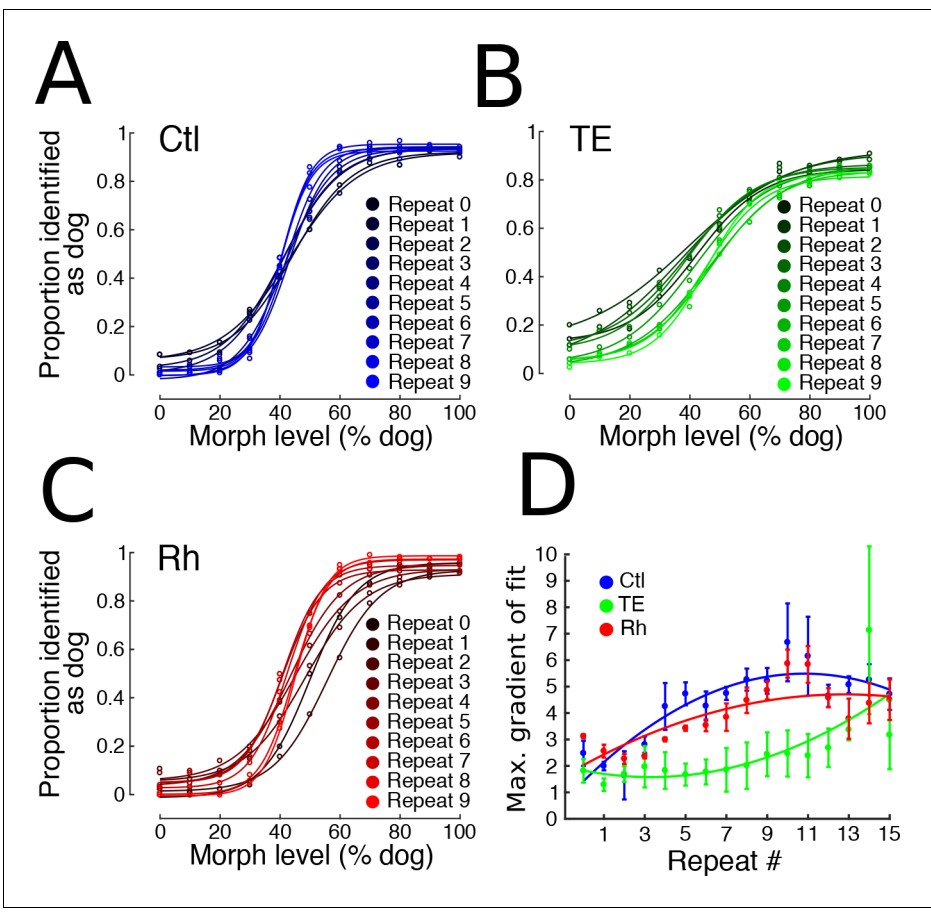

**Figure 2.** Experiment 1. (**A, B, C**) Categorization performance of control (n = 3), TE-lesioned (n = 3), and rhinal-lesioned (n = 3) groups, respectively, during the first 10 presentations of this stimulus set (10 of 16 total presentations plotted for clarity). A steeper gradient to the central portion of the sigmoid indicates higher classification accuracy. Data fit with the function: $a + b/(1 + exp(c * x + d))$, where $a$, $b$, $c$, and $d$ are free parameters. (**D**) The maximum slope (±s.e.m.) of the fitted functions in 1C, D and E plotted across presentations, fitted with a quadratic function for each group.

DOI: https://doi.org/10.7554/eLife.36310.004

The following source data is available for figure 2:

**Source data 1.** Experiment 1 - learning to categorize morphed images.

DOI: https://doi.org/10.7554/eLife.36310.005

(from 0% to 100% dog), but biased the distribution of stimuli towards the category boundary (cat 50:50 dog) by presenting stimuli consisting of the following ratios of cat:dog: 100:0, 75:25, 65:35, 60:40, 55:45, 50:50, 45:55, 40:60, 35:65, 25:75, and 0:100 (supplementary *Figure 1*). The asymmetrical reward structure of the task design produced a bias in all treatment groups toward classifying ambiguous stimuli as dogs, that is at the 50% morph level an unbiased subject would report the stimuli as dog-like 50% of the time. However, control monkeys reported the 50:50 morphs as more dog-like on 70% of presentations. The rhinal-lesioned group showed a similar bias, reporting 64% as more dog-like. The TE-lesioned group showed a bias in the same direction as the other two groups, reporting 50:50 morphs as more dog-like 61% of the time. For this experiment, we are interested in the ability to categorize visual stimuli accurately which is measured by the steepness of the discrimination curve. To remove the confound of learning rate, and focus on perceptual processing, we continued testing the three different treatment groups until performance reached asymptotic levels. For Experiment 2, all groups had reached asymptotic performance by the tenth presentation of the stimuli. During presentations 11 to 20, the categorization accuracy of monkeys with bilateral aspiration removals of rhinal cortex was indistinguishable from that of controls (LME, p=0.49, $z = -0.68$). Monkeys with bilateral TE removals made significantly more incorrect assignments than the other two groups (LME, p=$2.64 \times 10^{-10}$, $z = -6.32$), but nonetheless categorized better than would be expected by chance (t test, p=0.00057, d.f. = 5, $t = -7.49$) (*Figure 3A*). It seems unlikely that the impairment in making visual category discriminations following TE removals could be attributed to a learning deficit, because performance of all three test groups was stable throughout the latter 10 presentations of the test stimuli (LME, effect of repetition, p=0.54, $z = -0.61$).

For all monkeys, the closer the stimulus was to the category boundary, the longer it took to release the lever on correct trials during the presence of the red target (LME – effect of morph level, p=0.0017, d.f. = 5, $t = 5.23$) (*Figure 3B,C,D*), supporting the inference that classification is more difficult near the category boundary. The latency to release following the green target was constant across difficulty levels because the monkeys had presumably made the decision that the stimulus was dog-like earlier in the trial (LME – effect of morph level, p=0.40, d.f. = 5, $t = 0.26$). We interpret the response time to the green target as the basic visual-motor reaction time. Reaction times did not differ across groups, suggesting that TE removal did not slow the decision process (reaction times during red target (LME, p=0.15, d.f. = 2, $t = 1.35$)), nor did it slow the basic visual-motor reaction time (reaction time during green target (LME, p=0.48, d.f. = 2, $t = -0.069$)). The deficits observed in the TE group are also unlikely to be due to a failure in basic visual acuity, because grating contrast-sensitivity – a test designed to assess the visual acuity of human subjects (*Blakemore and Campbell, 1969*) - was indistinguishable across all three groups (Ctl vs TE: LME, p=0.28, d.f. = 2, $t = -0.69$; Ctl vs Rh: LME, p=0.22, d.f. = 2, $t = -0.95$)) (*Figure 4*), and similar to those of humans (*Blakemore and Campbell, 1969*).

Motivation and attention did not seem to be altered in either lesion group because the reaction times of all groups were indistinguishable, and there were few late bar release errors (<1% on average) – the few late release errors that were recorded were not distributed significantly differently among the treatment groups (Kruskal-Wallis, $\chi^2 = 2.49$, p=0.29).

## Experiment 3 – area TE impairment with visually degraded stimuli

Experiment 3 tested the possibility that monkeys with large cortical removals (TE/rhinal) had compensated for a deficit in object perception/categorization by memorizing one or more simple diagnostic features of each morph series (e.g. the 'tail' of the stimuli in the second row of *Figure 1C*). It has been demonstrated previously that the performance of monkeys with perirhinal cortex lesions is indistinguishable from controls in discriminating perceptually dissimilar stimuli obscured by masks (*Hampton and Murray, 2002*). The rationale for using the approach here, with perceptually similar stimuli, was if the monkeys were relying on a single diagnostic feature of the morphed stimuli to establish category membership in Experiment 2, image degradation of this nature (*Figure 5A*) would have a detrimental effect on performance. Consistent with this hypothesis, monkeys with bilateral TE removals were severely impaired by the masks relative to their own overall performance in the interleaved unmasked trials (*Figure 5D*) (LME, p=$2 \times 10^{-16}$, $z = 9.03$). They categorized the masked stimuli significantly less accurately than controls (*Figure 5B*) (LME, p=0.0017, $z = 3.13$). The performance of controls and that of monkeys with rhinal removals was indistinguishable (LME, p=0.20, $z = -1.27$); both groups exhibited smaller reductions in classification accuracy than the TE group in the presence

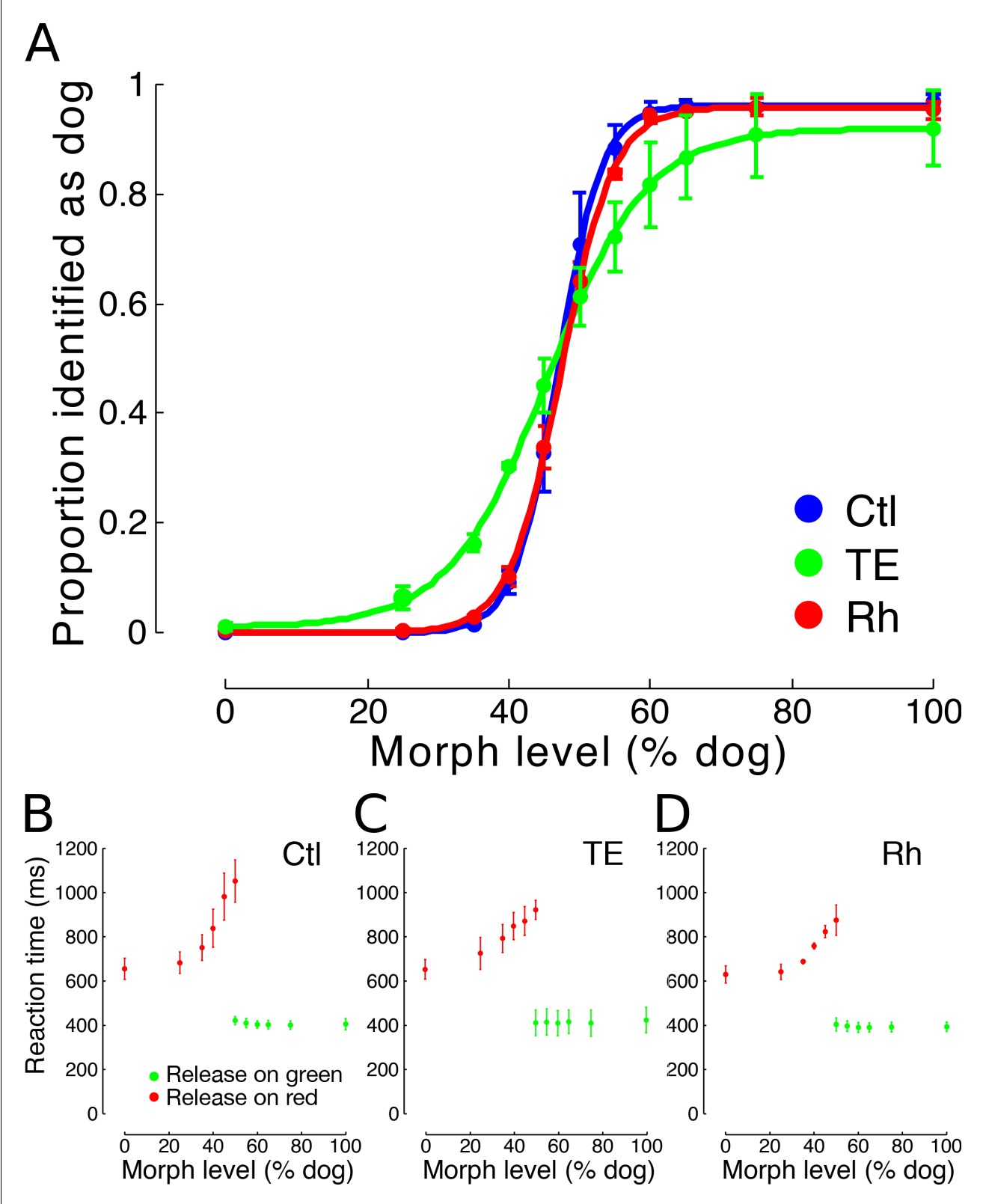

**Figure 3.** Experiment 2. (A) Categorization performance of three groups of monkeys: controls (n = 3), TE-lesioned group (n = 3), and Rh-lesioned group (n = 3), mean (±s.e.m.) of presentations 10 to 20. Data fit with the function: $a + b/(1 + \exp(c * x + d))$, where a, b, c, and d are free parameters. (C, D, E) Mean reaction times (±s.e.m.) of control, TE-lesioned, and rhinal-lesioned groups, respectively, for trials ending with a correct response. All trials at the 50% level are included. See figure supplement for stimulus examples.

*Figure 3 continued on next page*

*Figure 3 continued*

DOI: https://doi.org/10.7554/eLife.36310.006

The following source data and figure supplements are available for figure 3:

**Source data 1.** Experiment 2 - asymptotic categorization performance.

DOI: https://doi.org/10.7554/eLife.36310.009

**Figure supplement 1.** The cat-dog morphed images presented as visual stimuli in Experiment 2.

DOI: https://doi.org/10.7554/eLife.36310.007

**Figure supplement 2.** (A) Categorization performance of individual monkeys performing Experiment 2.

DOI: https://doi.org/10.7554/eLife.36310.008

of the masks, relative to the inter-leaved unmasked trials (*Figure 5C,E*) (Ctl: LME, p=$8.92\times10^{-11}$, $z = 6.48$; rhinal: LME, p=$1.33\times10^{-10}$, $z = 6.42$). Due to the asymmetrical reward structure of this and similar tasks, when monkeys fail to discriminate between the two offers they will usually resort to releasing the lever during the presence of the green target 100% of the time, as this is the only condition in which reward is available. This bias is visible in the impaired performance of the TE-lesioned monkeys following the foreground masking of the stimuli performed in Experiment 3.

## Experiment 4 – area TE impairment with a novel stimulus set

In Experiment 4, a large set of novel morphs were used as trial-unique stimuli in a single session to control for the possibility that the monkeys memorized individual stimulus-response outcomes in Experiments 1, 2 and 3. It has been demonstrated that visual discrimination of complex stimuli can be performed independently of area TE and of rhinal cortex if the stimuli have become sufficiently well learned (*Eacott et al., 1994*). We morphed a set of cat images each with two dog images, and vice versa (*Figure 6A*, supplementary *Figure 2*). This manipulation reduces the utility of a strategy focused on a single memorized feature (e.g. in *Figure 4A*, note the difference in appearance of the tail of 'Cat A' at the 50% morph level when paired with 'Dog A', versus the 50% level when paired with 'Dog B'). Consistent with the results of Experiment 3, monkeys lacking area TE categorized stimuli significantly less accurately than controls (LME, p=$2\times10^{-16}$, $z = 9.18$), whereas those with rhinal cortex removals were indistinguishable from controls (*Figure 6B*) (LME, p=$-0.62$, $z = -0.50$).

## Discussion

According to the hierarchical model of object perception, area TE plays an important role in the integration of visual features into identifiable objects. The results from the experiments described above suggest that area TE is only important for perceiving and classifying objects with complex over-lapping features, especially when novel/unfamiliar. The monkeys with TE removals were surprisingly good at classifying all but the most ambiguous stimuli (i.e. those closest to the category boundary). Perception of simple, low-level features appears to be intact in the monkeys lacking TE as shown by their normal ability to detect the change in color of the target from red to green in Experiments 1 – 4, and by their normal contrast sensitivity when tested with black and white sine wave gratings.

There is a debate in both human and non-human primate literature about whether medial temporal lobe tissue - rhinal cortex - is important for the perception of feature-ambiguous stimuli (*Lee et al., 2005*; *Levy et al., 2005*; *Buckley and Gaffan, 2006*; *Baxter, 2009*; *Suzuki, 2009*; *Lee and Rudebeck, 2010*). The proposal that the visual object processing hierarchy includes perirhinal cortex arose from studies claiming a role for monkey perirhinal cortex in visual perception following tests of visual discrimination, oddity detection, or delayed-match-to-sample (DMS) at zero delay (*Eacott et al., 1994*; *Buckley et al., 2001*; *Bussey et al., 2003*). The aforementioned tasks required monkeys to compare an image with one or more specific images that must be maintained in memory (even in the case of the oddity task – in which stimuli appear simultaneously – the subject must maintain a representation of two or more features from three or more images to make a correct choice, a process that presumably requires the subjects to look at each image in turn and remember those already looked at). In 'delayed matching' tasks, monkeys with removals of rhinal cortex are only impaired when there is a delay of several seconds between the sample and test stimuli (*Buffalo et al., 1999*), and have impaired short-term memory but intact habit formation for the exact same images (*Tu et al., 2011*). In the present study, monkeys that had received lesions several years

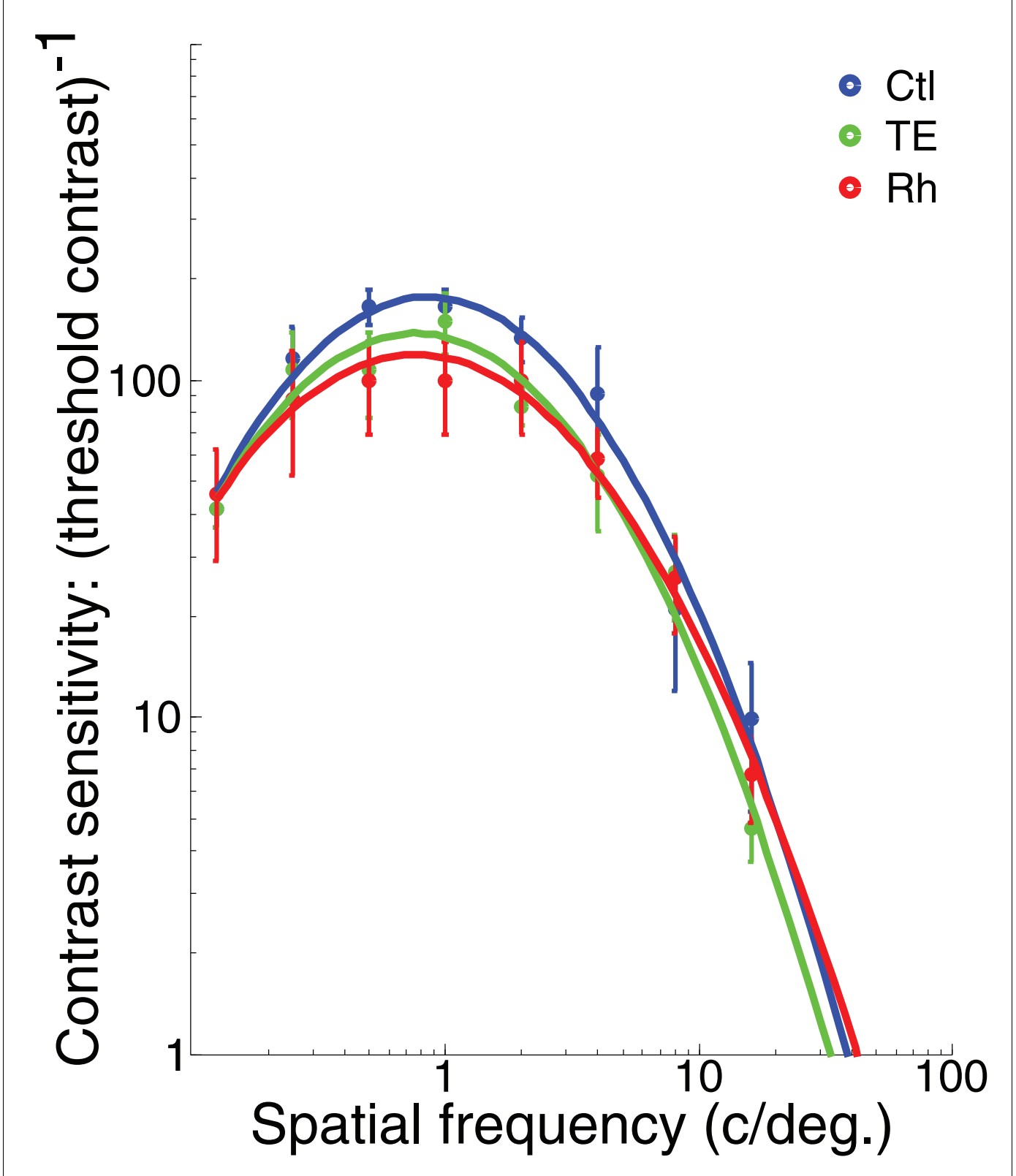

**Figure 4.** Visual acuity testing. Contrast sensitivity is plotted on a logarithmic scale against spatial frequency. Mean sensitivity (±s.e.m.) for each of the three groups of monkeys - controls (n = 3), TE-lesioned group (n = 3), and Rh-lesioned group (n = 3), (six technical replicates per monkey) - is fit with a quadratic function.

DOI: https://doi.org/10.7554/eLife.36310.010

*Figure 4 continued on next page*

*Figure 4 continued*

The following source data is available for figure 4:

**Source data 1.** Visual acuity testing.

DOI: https://doi.org/10.7554/eLife.36310.011

prior to testing (and hence compensatory reorganization in response to the testing series is unlikely) performed a task which only requires memory for a categorical exemplar or boundary, along with the category-response mapping. The anatomical extent of the rhinal removals was comparable to those described in the above-mentioned studies. Nonetheless, performance of rhinal-lesioned monkeys was indistinguishable from that of control monkeys by all measures. It could be argued that because categorization tasks require generalization across perceptually similar stimuli, that such tasks may be solved using features of intermediate complexity, rather than gestalt object representations (the latter being the type of perceptual function some have tried to ascribe to rhinal cortex (*Bussey et al., 2003*)). Although we cannot preclude this possibility, the large degree of feature overlap in our stimulus sets (c.f. the 40% vs. 60% morphed stimuli in *Figure 1C*) means that there are no obvious such intermediate features on which the task could be solved. When the discriminations are made more difficult (Experiments 3 and 4), there are no large changes in the response selections of the control and rhinal-lesioned groups. However, there are two changes in the response patterns of the TE-lesioned monkeys; first, their discrimination performance becomes considerably worse as shown by the decrease in slope of the discrimination curve. Second, the monkeys have changed their criterion for classifying a stimulus as a dog as shown by the shift in the 50% discrimination point to the left. Since these changes occur when the discrimination is made difficult, we can conclude that the monkeys have a discrimination deficit, and in the face of the discrimination being made more difficult the monkeys have changed their criterion by classifying more trials as dogs. The change in criterion is not surprising in the face of the asymmetrical reward structure of the task. The most parsimonious interpretation of these observations is that monkeys assess each image in its entirety in order to make a categorical judgement, and hence it seems that the TE-rhinal boundary represents the end of perceptual processing in the ventral visual stream.

The lack of evidence for a role of rhinal cortex in perceptual processing reported here suggests that previous reports of deficits in visually guided behavior following rhinal removals may be attributable to memory deficits. This interpretation is supported by a number of studies demonstrating a role for rhinal cortex in mnemonic processes. In behavioral testing, monkeys lacking rhinal cortex are impaired in object recognition (*Meunier et al., 1993*), and learning stimulus-stimulus associations (*Murray et al., 1993*). They are also severely impaired in comparing reward values across time (*Liu et al., 2000*), even when vision plays no role in the task (*Clark et al., 2012*). Rhinal cortex neurons appear to be tuned for complex stimuli in a similar manner to those recorded in area TE. However, neuronal responses in TE generally exhibit task-invariant tuning for specific stimuli (*McMahon et al., 2014*), whereas neuronal responses in rhinal cortex are modulated by behavioral context (*Yakovlev et al., 1998*; *Lehky and Tanaka, 2007*; *Naya and Suzuki, 2011*; *Eradath et al., 2015*). For example, when the stimulus-mapping in a delay-discounting task is randomized, perirhinal neurons lose their selectivity on the next trial (*Liu and Richmond, 2000*). The loss of stimulus selectivity in perirhinal neurons following a contextual change implies that the apparent selectivity is derived from a learned association with predicted outcome.

Rhinal cortex neurons are also sensitive to learned stimulus-stimulus associations; they form a functional microcircuit that is dynamically modulated by task demand (*Erickson and Desimone, 1999*; *Hirabayashi et al., 2013*) and they convey information relevant to behavioral choice in a more accessible manner than those in TE (*Pagan et al., 2013*; *Pagan and Rust, 2014*). Furthermore, the delay between TE and perirhinal neural responses to visual stimuli is considerably longer (~60 ms) than is usually attributed to a direct feed-forward mechanism (10–15 ms) (*Xiang and Brown, 1998*; *Liu and Richmond, 2000*); this suggests that rhinal cortex is doing more than a simple monosynaptic transformation of visual input from TE neurons. In the current study, a variety of manipulations that increased the perceptual difficulty of a categorization task revealed significant, although far from catastrophic, deficits in monkeys with bilateral TE removals, and no deficits in monkeys with bilateral rhinal removals. Thus, the final stage of image processing in the ventral visual stream appears to

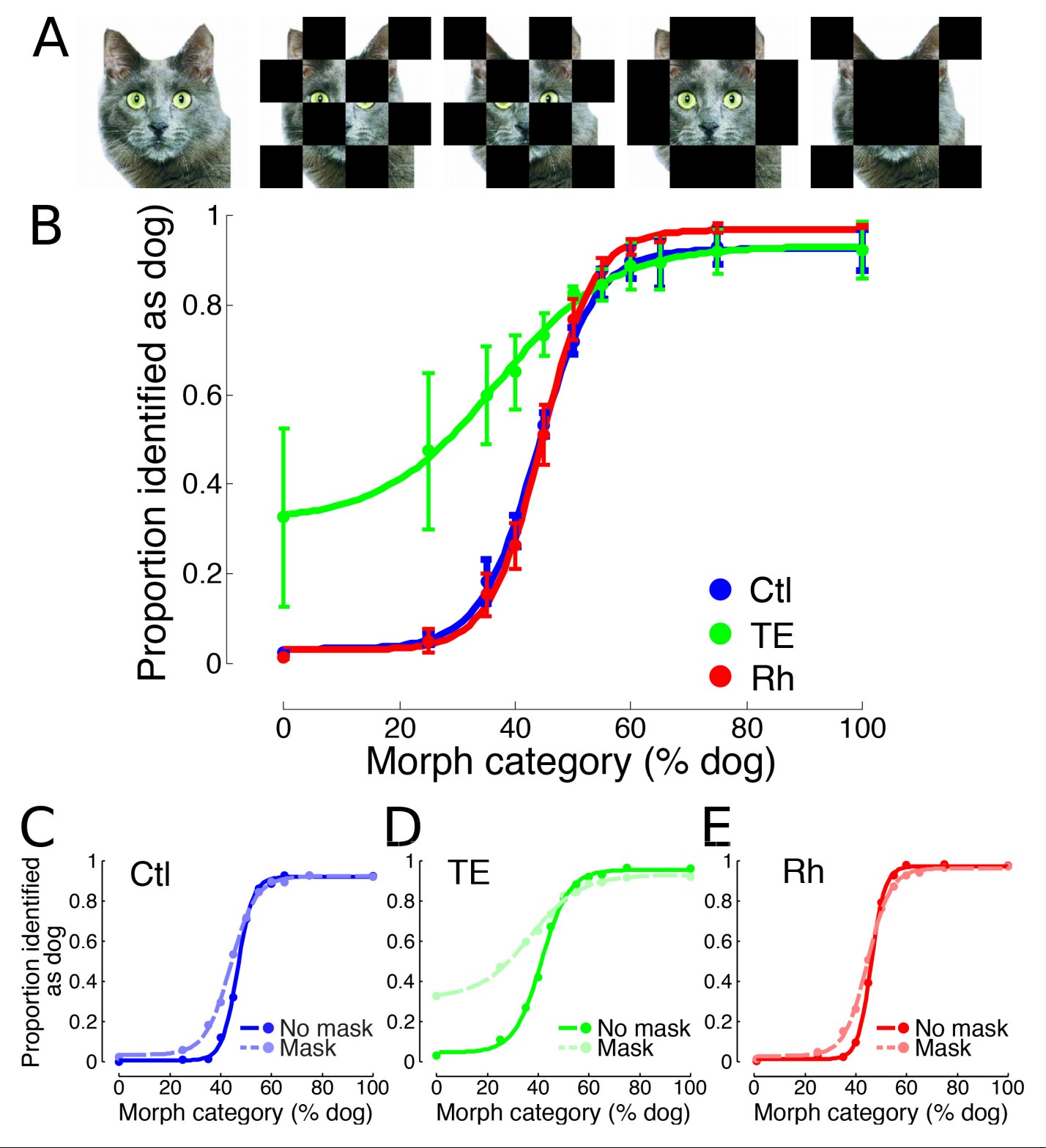

**Figure 5.** Experiment 3. (**A**) Examples of the visual stimuli presented. Four checker-board masks were placed over each of the stimuli used in Experiment 1, and presented inter-leaved with an unmasked version of each stimulus. (**B**) Categorization performance of the three test groups: mean (±s.e.m.) of responses to first presentation of all masked stimuli. (**C, D, E**) Categorization performance on masked (mean of all masks) vs. unmasked stimuli for each group, respectively (first presentation).

DOI: https://doi.org/10.7554/eLife.36310.012

*Figure 5 continued*

The following source data and figure supplement are available for figure 5:

**Source data 1.** Experiment 3 - categorization of visually degraded stimuli.
DOI: https://doi.org/10.7554/eLife.36310.014
**Figure supplement 1.** (A, C, E) Categorization performance on unmasked stimuli presented in Experiment 3, for individual control (A), TE-lesioned (C), and rhinal-lesioned (E) monkeys.
DOI: https://doi.org/10.7554/eLife.36310.013

occur in area TE. Rhinal cortex is critical for learning and remembering contextual or associative relations among stimuli or events, but appears to play no role in object percept formation.

## Materials and methods

### Subjects and surgeries

All experimental procedures conformed to the *Institute of Medicine Guide for the Care and Use of Laboratory Animals* and were performed under an Animal Study Proposal approved by the Animal Care and Use Committee of the National Institute of Mental Health. Subjects were nine adult male monkeys (*Macaca mulatta*). Three monkeys (5 – 6 years old, weighing 6.9 – 9.0 kg) had previously received bilateral aspiration removals of area TE 2 years prior to the commencement of this study; the reconstructions of these lesions have been published previously (*Matsumoto et al., 2016*). Three monkeys (7 years old, weighing 7.0 – 14.5 kg) received bilateral aspiration removals of rhinal cortex (Rh) (comprising peri- and ento-rhinal cortex - comprising areas 28, 35, and 36 of Brodmann) 3 years prior to this study. Aspiration removals of rhinal cortex have been described previously (*Meunier et al., 1993*; *Fritz et al., 2005*); the Rh removals were largely as intended (supplementary *Figure 3*). Three monkeys (8 – 11 years old, weighing 7.8 – 9.5 kg) were unoperated controls.

### Behavior

Monkeys sat in a primate chair inside a darkened, sound-attenuated testing chamber. They were positioned 57 cm from a computer monitor (Samsung 2233RZ) (*Wang and Nikolić, 2011*) subtending 40° × 30° of visual angle. Task timing and visual stimulus presentation were under the control of networked computers running, respectively, custom written (Real-time Experimentation and Control, REX (*Hays et al., 1982*)) and commercially available (Presentation, Neurobehavioral Systems) software for the design and control of behavioral experiments. Monkeys were initially trained to grasp and release a touch-sensitive bar to earn fluid rewards. After this initial shaping, a red/green color discrimination task was introduced (*Bowman et al., 1996*). Red/green trials began with a bar press, 100 ms later a small red target square (0.5°) was presented at the center of the display (over-laying a white noise background). Animals were required to continue grasping the touch bar until the color of the target square changed from red to green. Color changes occurred randomly 500 – 1,500 ms after bar touch. Rewards were delivered if the bar was released between 200 and 1000 ms after the color change; releases occurring either before or after this epoch were counted as errors. Thus, the color change occurs randomly within a 1000 ms time window, and the behavioral response must be made within 1000 ms of the color change; this design encourages the monkeys to use the color target to guide their responses. A strategy based on timing alone would result in a theoretical maximum of less than 50% correct. All correct responses were followed by visual feedback (target square color changed to blue) after bar release and reward delivery 200 – 400 ms after visual feedback. There was a 2-s inter-trial interval (ITI), regardless of the outcome of the previous trial.

After an animal reached criterion in the red/green task (2 consecutive days with >85% correct performance) a visual discrimination task was introduced. Each trial began when the animal grasped the touch bar; bar press was now initially followed (after 100 ms) by the presentation of a cue image. For training, the cues were two black and white block ('Walsh') patterns (13° x 13°). The cues signaled whether a release during the green target would result in the delivery of a drop of liquid reward, or a 4000 – 6000 ms 'time-out'. The red target appeared 500 ms after the cue and changed color to green 2000 – 3000 ms later if the monkey continued to hold the bar. Monkeys could avoid the predicted outcome by releasing the lever before the red target transitioned to green; a new trial

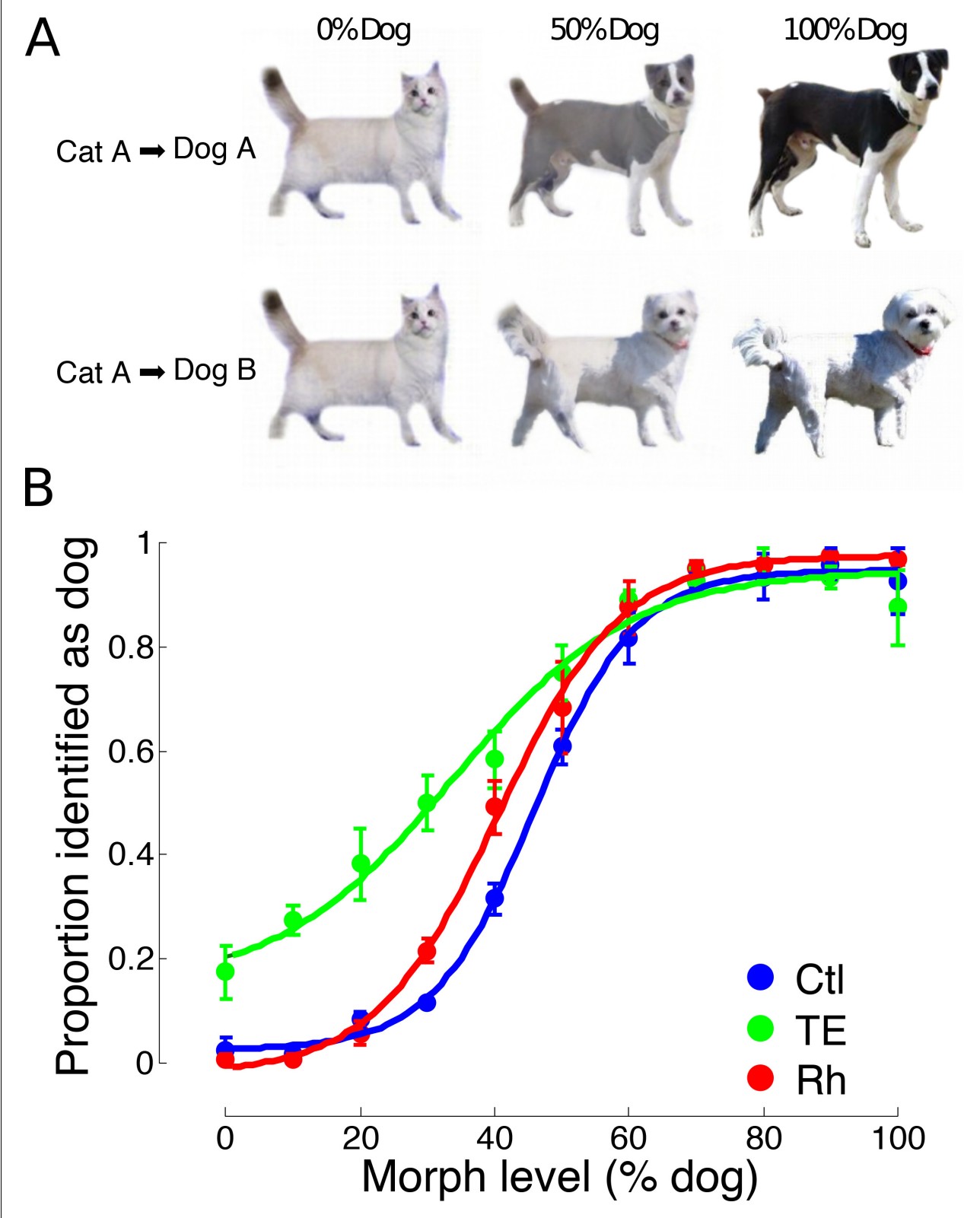

**Figure 6.** Experiment 4. (**A**) Examples of the visual stimuli presented; each cat was morphed with two dogs, and vice versa, for example Cat A was morphed with Dog A (top row), and with Dog B (bottom row). Examples at the 0%, 50%, and 100% dog level are shown; the full set of stimuli used in Experiment 4 was distributed across the same morph levels as used in Experiment 1 (see figure supplement for a larger set of stimulus examples). (**B**) Mean categorization performance (±s.e.m.) of the three test groups with a single presentation of each stimulus.

*Figure 6 continued on next page*

*Figure 6 continued*

DOI: https://doi.org/10.7554/eLife.36310.015

The following source data and figure supplements are available for figure 6:

**Source data 1.** Experiment 4 - categorization of novel stimuli.
DOI: https://doi.org/10.7554/eLife.36310.018
**Figure supplement 1.** Examples of the cat-dog morphed images presented as visual stimuli in Experiment 4.
DOI: https://doi.org/10.7554/eLife.36310.016
**Figure supplement 2.** Categorization performance of individual monkeys in Experiment 4, in which each stimulus was novel, and presented only once.
DOI: https://doi.org/10.7554/eLife.36310.017

could then be initiated after the standard ITI. After the monkeys became acclimated to the incentive difference between the two black and white cues (2 consecutive days with >85% correct performance), they progressed to category training. For category training, the two black and white cues were replaced with two sets of cues, that is 20 dogs as the rewarded set, and 20 cats as the unrewarded set. If the monkey released the lever during the green target when a dog was present, the monkeys received one drop of liquid reward (*Figure 1B*). If the monkey released during the green target when a cat was present, no reward was delivered and there was a 4 – 6 s time-out. There was never a reward for releasing while the red target was present. The optimal behavior is to release during the presentation of the red target for the trials on which cats are presented, essentially skipping on to the next randomly selected trial, and release during the presentation of the green target for the trials on which dogs are presented to obtain the reward. This design is effectively a visually cued two-interval forced choice (2- IFC) task, with asymmetrical reward, in which the color of the central target indicates the current 'choice window'. In the second phase of category training, monkeys were presented with four larger sets of trial-unique images (240 cats and 240 dogs), to confirm that the monkeys were able to classify stimuli based on visual perceptual categorization.

For the experiments with morphed stimuli, releasing the lever during the green target resulted in a 4 – 6 s time-out if the stimulus was more cat-like (i.e. <50% dog), and a reward if the stimulus was more dog-like (i.e. >50% dog). The outcome of trials on which a stimulus at the category boundary (i.e. == 50% dog) was presented was determined probabilistically; 50% of trials resulted in reward delivery, 50% resulted in a time-out. In Experiments 1 and 2, the stimulus set was presented twice each day, over eight and 10 days, respectively. In Experiment 3, the stimulus set was presented once over 2 days; with morph level and mask type counter-balanced across both days. The addition of the masked stimuli resulted in the set size for Experiment three being considerably larger than those used in the other experiments; testing was thus split over two sessions to ensure that the monkeys did not become sated or inattentive during performance. In Experiment 4, the stimulus set was presented once on a single day.

## Visual cues

All visual cues were jpeg or pcx format photos (200 × 200 pixels). The training sets of dogs/cats used in this study are the same as in our previous report (*Minamimoto et al., 2010*). The images used in the four experiments were generated from a subset of the training images, in which pairs of cats and dogs were used to create cat-dog morph sequences using Fantamorph software (Abrosoft, Beijing, China). For Experiment 1, 20 dogs were morphed with 20 cats. From each cat-dog morph series, a set of stimuli comprising cats, dogs, and intermediate morphed images, spaced at 10% increments from 0% to 100% dog, was derived (*Figure 1C*). For Experiment 2, the same cat-dog morph series were used as in Experiment 1, but the distribution of stimuli was concentrated around the category boundary (cat 50:50 dog); stimuli consisted of the following ratios of cat:dog: 100:0, 75:25, 65:35, 60:40, 55:45, 50:50, 45:55, 40:60, 35:65, 25:75, and 0:100 (*Figure 3*, *Figure 3—figure supplement 1*). For Experiment 3, the stimuli from Experiment two were presented; on four fifths of trials, the stimuli were overlaid with one of four coarse black-block masks (*Figure 5A*). For Experiment 4, a new set of 20 cats and 20 dogs was used to create cat-dog morph series. Each cat was morphed to each of two dogs, and vice versa, for a total of 40 cat-dog morph series, to make it more difficult to discriminate among images from a morph series based on a single prominent feature (*Figure 6A and 2*).

## Data analysis

Modeling and statistical methods were implemented in MatLab (MathWorks) and R (R core team, 2017). Linear mixed effects analyses (lme4; *Bates et al., 2015*) were used to evaluate the relationship between categorization accuracy (recorded as the proportion of trials classified as 'dog') and treatment group (control, TE, or Rh). The fixed effects entered into the model were treatment group, morph level, and, where relevant, repetition (with interaction terms). As random effects, we included the intercept for subject. This approach assumes that variation in repeated measures data is due to both fixed (for example, treatment group) and random (for example, monkey) effects, allows independent variables to be treated as continuous (for example, morph level) or categorical (for example, treatment group), and allows for non-normal dependent measures (that is, categorization accuracy). The data followed a logistic distribution, so we used a binomial link function.

## Acknowledgements

We thank Alex Cummins for histology support, Grace Mammarella for lesion reconstructions, and Adin Horowitz for assistance with behavioral testing. We are grateful to Drs Alex Martin, Chris Baker and Christian Quaia for comments on the draft manuscript. This work was supported by the Intramural Research Program, National Institute of Mental Health, National Institutes of Health, Department of Health and Human Services. The opinions expressed in this article are the authors' own and do not necessarily reflect the views of the US National Institutes of Health, the Department of Health and Human Services, or the United States Government.

## Additional information

### Funding

| Funder | Grant reference number | Author |
| --- | --- | --- |
| National Institute of Mental Health | 1ZIAMH002032-41 | Mark A G Eldridge<br>Narihisa Matsumoto<br>John H Wittig Jnr<br>Evan C Masseau<br>Richard C Saunders<br>Barry J Richmond |

The funders had no role in study design, data collection and interpretation, or the decision to submit the work for publication.

### Author contributions

Mark AG Eldridge, Conceptualization, Data curation, Software, Formal analysis, Supervision, Validation, Investigation, Visualization, Methodology, Writing—original draft, Project administration, Writing—review and editing; Narihisa Matsumoto, Conceptualization, Software, Formal analysis, Validation, Investigation, Visualization, Methodology, Writing—original draft, Writing—review and editing; John H Wittig Jnr, Conceptualization, Resources, Data curation, Software, Supervision, Investigation, Methodology, Writing—original draft, Writing—review and editing; Evan C Masseau, Conceptualization, Resources, Data curation, Software, Investigation, Methodology, Project administration; Richard C Saunders, Conceptualization, Resources, Formal analysis, Supervision, Funding acquisition, Investigation, Visualization, Methodology, Writing—original draft, Project administration, Writing—review and editing; Barry J Richmond, Conceptualization, Resources, Data curation, Software, Formal analysis, Supervision, Funding acquisition, Validation, Investigation, Visualization, Methodology, Writing—original draft, Project administration, Writing—review and editing

### Author ORCIDs

Mark AG Eldridge http://orcid.org/0000-0003-4292-6832
John H Wittig Jnr http://orcid.org/0000-0003-0465-1022
Barry J Richmond http://orcid.org/0000-0002-8234-1540

## Ethics

Animal experimentation: All experimental procedures conformed to the Institute of Medicine Guide for the Care and Use of Laboratory Animals and were performed under an Animal Study Protocol approved by the Animal Care and Use Committee of the National Institute of Mental Health, covered by project number: MH002032.

## Decision letter and Author response

Decision letter https://doi.org/10.7554/eLife.36310.021
Author response https://doi.org/10.7554/eLife.36310.022

## Additional files

### Supplementary files

• Transparent reporting form
DOI: https://doi.org/10.7554/eLife.36310.019

### Data availability

All data generated or analysed during this study are included in the manuscript and supporting files.

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
