## [Decision Letter]

Thank you for submitting your article "Where object perception ends in the ventral visual stream" for consideration by *eLife*. Your article has been reviewed by three peer reviewers, and the evaluation has been overseen by a Reviewing Editor and David Van Essen as the Senior Editor. The following individual involved in review of your submission has agreed to reveal his identity: Robert Hampton (Reviewer #2).

The reviewers have discussed the reviews with one another and the Reviewing Editor has drafted this decision to help you prepare a revised submission.

Summary:

In this manuscript, Eldridge et al. investigate whether the rhinal cortex is involved in visual perception or whether upstream visual area TE is the "terminal" stage in visual processing in the ventral visual stream. To answer this question, the authors compared and contrasted behavioral performance of groups of monkeys with bilateral rhinal cortex lesions, bilateral TE lesions and unoperated controls. They employed a basic two-interval, forced choice, cat-dog categorization task in which monkeys were asymmetrically rewarded for releasing a lever during a "choice window" indicated by the color of the central target (red target for cat-like stimuli and green target for dog-like stimuli). The stimulus set consisted of cats, dogs, and intermediate morphs between the two. The authors find that while all groups of monkeys classified most stimuli well on the first repetition, the rates of improvement towards asymptotic performance were variable in different groups. While the TE lesioned monkeys improved the slowest over repetitions, the rhinal lesioned monkeys and the unoperated controls improved at a faster rate and did not differ from each other. Further, the authors used modified versions of the task to test the robustness of their results in the face of task difficulty. In sum, the authors argue that while area TE is required for visual perception, rhinal cortex is not necessary. This is taken to contribute to the ongoing debate about the role of perirhinal cortex in perception and suggest that area TE marks the end of purely visual processing in the ventral visual stream and does not extend to anatomically adjacent rhinal cortex.

Essential revisions:

Overall, this is an important study which is expected to be of interest to a broad array of visual, cognitive and behavioral neuroscientists. At the same time, however, there are substantial concerns about confounding factors in the task design and behavior that need to be discussed and addressed. In addition, improvements and additional analyses are needed for a more rigorous evaluation and a more complete picture of the issues under study.

1) Task-design:

1a) In the two-interval forced choice, cat/dog categorization task, is the first choice target always red and the second choice target always green? Or is the order of these two colored targets counterbalanced? In other words, is the first interval always correlated with the red target for cat stimuli and the second interval always correlated with green target for dog stimuli? Are monkeys using color to do the task? Or are they relying on an unknown combination of color and interval, so that it is not possible to know for sure what the monkeys are relying on since they are correlated?

1b) Clarification in text: In the subsection “Experiment 1 – learning to categorize morphed stimuli”, where the task is described, it is stated that monkeys were rewarded for responding with a lever release for cat-like stimuli in the red (first) interval and for dog-like stimuli in the green (second) interval. However, in the second paragraph of the subsection “Behavior”, it states that only responses for dog-like stimuli during the green (second) interval were rewarded and there was never a reward for releasing while the red target was present. There is a discrepancy here and needs to be resolved.

2) Psychophysical results (Figures 2, 5 and 6):

If the task contains an asymmetrical reward structure in which monkeys are rewarded only for the "dog" category, can the results in Figure 2B be explained by this reward bias? It looks like psychophysical performance for TE lesioned monkeys is impaired only for 0-20% dog stimuli (which are essentially cat-like stimuli). It is possible that monkeys are very good at the category that is rewarded and not good for the category that is unrewarded. Bias in performance is also present only for TE lesioned monkeys and not for controls or rhinal lesioned monkeys. Further, in Figures 5 (5B and 5D) and 6 (6B), the patterns of deficits in TE monkeys is biased towards the left of the psychophysical curve (just like in Figure 2). More insight is needed to understand whether this bias is a result of the asymmetrical reward structure? Does the bias switch if only cat stimuli were rewarded?

Also, for TE lesioned monkeys (Figure 2B), saturating performance (at 80-100% morph levels) decreases with repetition but not for controls or rhinal lesioned monkeys. Is this statistically significant?

3) Behavior and training for Figure 2:

What was the frequency of repetitions for the three groups of monkeys? Was it one repetition per day and was it similar across all the individual monkeys in the three groups? Could the impaired learning in TE lesioned monkeys be a result of more repetitions within the same time interval for control and rhinal lesioned monkeys but lesser repetitions within the same time interval for TE lesioned monkeys?

4) Analysis for Figure 3:

The text (subsection “Experiment 2 – area TE removal impairs perceptually difficult categorization”, first paragraph) mentions that the analysis used only sessions 11-20 in which asymptotic performance was reached to remove the effect of learning. However, in the subsection “Experiment 1 – learning to categorize morphed stimuli”, it states that while controls and rhinal lesioned monkeys reached asymptotic performance in 10 sessions, TE lesioned monkeys reached asymptotic performance in 14 sessions. Yet, to compare across groups for Figure 3, sessions 11-20 have been used for analysis, even though TE lesioned monkeys did not reach asymptotic performance by session 11. It might be cleaner to report analysis for Figure 3 using sessions in which all monkey groups reached asymptotic performance (perhaps session 14-20). The results in Figure 3A could be attributable to slower learning in TE.

5) Reaction-time distributions in Figure 3:

In Figure 2C, the variance in RTs for release on red is lower than the variance in RTs for release on red in Figure 2B. Also, in Figure 2C, the variance in RTs for release on green is higher than the variance in RTs for release on green in Figure 2B and 2D. In both the lesioned monkey groups (2C and 2D), RTs for release on red are faster on the hardest conditions (near boundary) as compared to controls. Is there an explanation for these effects?

6) Color vision related to Figure 4:

Previous studies (Buckley 1997) have shown that TE lesions cause deficits in color discrimination. Do the anatomical locations of TE lesions in this study match with other studies that show color deficits? Does the TE lesioned group have difficulty in discriminating color? And does this affect their performance on the task? In the first paragraph of the Discussion, it mentions that these monkeys can detect the change from red to green. Is this from a different task? This should be explained in the text or referenced in some form.

7) How does the size of the TE lesion compare to the size of the rhinal lesion? For example, if TE lesions are larger than rhinal lesions, this could potentially lead to more deficits in TE lesioned groups than rhinal lesioned groups.

8) In both groups of lesioned monkeys, is there a reduction in attention or motivation during task performance? For instance, are more trials aborted before completion or are there more fixation breaks – and do these differ between monkey groups?

9) Please include plots of the data from individual TE animals for all of the figures where a deficit is shown (in order to evaluate the heterogeneity of performance, and to see in which animals a reward bias could account for their behavior). Without seeing the TE lesions (including subcortical structures), we cannot know if they encroached upon reward circuitry or other brain regions to a greater extent than the Rh lesions, allowing non-perceptual explanations of the deficit to remain possible.

10) Please clarify how the present task differs from Task 1 of Lee et al., 2005, and how it has reduced memory demands relative to Lee and Rudebeck, 2010. There is a large prior literature showing Rh lesion-induced perceptual deficits that may be prematurely dismissed on the basis that those tasks did not eliminate memory demands adequately. But those tasks were described as having almost non-existent memory demands. Lee and Rudebeck, 2010, used a "possible/impossible" object judgement with no memory demands. Similarly, Lee et al., 2005 used a task almost identical to the present monkey task except that it displayed 2 stimuli simultaneously, with an identical category judgment (referring to a 'target' category defined at task outset); thus, in Lee et al., the greatest memory demand was comparing to a target category held in mind (as in the present study), not comparing between two adjacent stimuli. Second, it seems implausible that the extremely short memory demands imposed by prior oddity tasks (saccading between adjacent stimuli) are enough to render Rh involvement critical. This implies a role for Rh cortex in sensory/perceptual integration so extreme that patients with Rh damage should be unable to make sense of the dynamic visual world. Please provide further discussion of this important point. You may not be able to fully resolve this as it will likely remain hotly debated but describing your specific concerns in a convincing and detailed way will go much farther to increase the impact of this paper.

[Editors' note: further revisions were requested prior to acceptance, as described below.]

Thank you for resubmitting your work entitled "Perceptual processing in the ventral visual stream requires area TE but not rhinal cortex" for further consideration at *eLife*. Your revised article has been favorably evaluated by Eve Marder as the Senior Editor, a Reviewing Editor, and three reviewers.

This revision was very responsive in addressing several questions posed by the reviewers and providing further clarification of the design and interpretation of the reported results. There was not complete agreement across the reviewers as reviewer 1 suggests an intriguing alternative interpretation for the results that we hope will be the basis for further discussion. Their re-review is included below and we would ask that you consider their comments and make some final revisions to your paper to respond to their concerns.

Reviewer #1:

I remain unconvinced that the results and their interpretation are sufficiently compelling for publication in *eLife* (but I accepted the earlier, collective "letter to the authors" that was reasonably positive, because I see that I might be in the minority). My reservations stem from the ambiguous nature of the task (and the pattern of results in individual TE monkeys) and from weaknesses in the arguments for the key interpretations/conclusions. I respond to the authors' response letter below.

Authors: We have included plots of individual monkey performance for all experiments in which a deficit was reported (Figure 3—figure supplement 2; Figure 5—figure supplement 1; Figure 6—figure supplement 2).

Reviewer: This confirms what I ascertained by looking at the raw data: the TE lesion effect comes almost entirely from one monkey (K). In the other 2 TE monkeys, to the extent that they show any deficit, it is in classifying cats as dogs without a symmetrical tendency to classify dogs as cats. In other words, in these two monkeys, their small number of errors could be due to response bias (greater propensity to release the bar on green, because of increased reward-seeking/reduced risk-aversion, etc.). The authors state in the revised manuscript that "Due to the asymmetrical reward structure of this and similar tasks, when monkeys fail to discriminate between the two offers they will usually resort to releasing the lever during the presence of the green target 100% of the time, as this is the only condition in which reward is available." This illustrates the ambiguous nature of the task: if monkeys show a bias toward releasing on green (i.e. selecting "dog") it could be due to failure to discriminate or due to response bias (since the assignment of reward contingencies is not symmetric with respect to cat/dog category) and we cannot know which. This weakness is a key reason why I do not feel the present results are compelling enough to warrant publication in *eLife*.

Authors: Lee et al., 2001, 'task one' is a classic visual discrimination task (simultaneous S+ vs. S-), where the categories offer no information that can help the subjects solve the task.… Lee 'task 2' is something of a hybrid between visual discrimination and oddity tasks; the subject can use one stimulus to determine which of the other two stimuli is most perceptually similar (i.e. the target), and thus must make real-time comparisons among the 3 stimuli presented (further discussion of this point below). 'Task 2' could also be solved using the same visual discrimination learning required in 'task 1', although we agree with the authors that it is parsimonious to assume that the adoption of the former strategy is more likely. In both tasks, the subjects are being asked to make a stimulus-reward association – task 1 over the longer timescale of multiple presentations; task 2 over the shorter timescale of within-trial presentations. Our task places no similar requirement for 'stimulus-reward' mapping on our subjects. The conclusion we draw from these comparisons is that Rh cortex is likely critical for stimulus-reward association memory (as others have demonstrated), but that memory for 'category' is supported elsewhere (earlier in the visual system).

Reviewer: The authors argue that the key distinction between the Lee et al. tasks and their task is the use of only a single item per learned discrimination (in Lee et al.) versus the use of *sets* of items constituting the to-be-discriminated categories (i.e. multiple cats and multiple dogs) in their task. In Lee et al., the requirement to associate each single item with reward thus renders the task a "memory" task. Whereas, in the authors' task, because monkeys had to generalize across multiple items within a category and correctly associate that category with reward, the task is not a memory task but a perceptual discrimination task. Note that "and correctly associate that with reward" was added by me – the authors make no mention of reward in describing their own task. Yet an association between the category and the reward is critical for correct performance in the authors' task, and in fact this category-to-reward mapping is more complex than in the Lee et al. tasks (monkeys must learn: if "dog", then release while green; if "cat", then do not release until red). To argue that a simpler reward contingency (in Lee et al.) makes the task more "mnemonic" than a complex reward contingency (in the present task) seems backwards. I agree that what makes the present task different from the Lee et al. tasks is the requirement for generalizing across category exemplars, but this implies a very different interpretation than the one offered by the authors. The most sensible interpretation is as follows. The present task is a categorization task that requires both generalization across diverse cats (or across diverse dogs) *and* discrimination of cats from dogs. Therefore, the optimal representations are not whole, unique objects (residing in rhinal cortex), but rather Shimon Ullmann-esque intermediate complexity features (known to reside in IT) that allow generalization across distinct cats as well as discrimination of cats from dogs. This accords with another interesting/complicating factor in the present task: owing to the distribution of perceptual features possessed by cats and dogs, most cats are a plausible subset of dogs, but most dogs are not a plausible subset of cats (Mareschal, French and Quinn, 2000; Mareschal, Quinn and French, 2002). Given this category asymmetry, having compromised IT representations (needed for generalization and discrimination) might lead to a bias toward classing cats as dogs, as seen in the data. One way to test this would be to replicate the study in a design without reward asymmetry, to see if the dog-bias (which could then only arise from inherent category asymmetry) still exists.

Authors: Lee and Rudebeck, 2010, implemented a task that required subjects to report whether drawings of stimuli were viable as 3D objects. The memory demands of that task are much more similar to those of the present study – the subjects classified stimuli into one of two categories. However, the task only tests 'perception' in an abstract sense; there remains a confound with cognitive load – the mental reconstruction of a 3-dimensional image from a 2-dimensional representation demands more than simple perception of the object as a whole.

Reviewer: Here, it is not clear how the authors' argument rebuts the reviewer's concern. Do the authors mean to equate "mental reconstruction of a 3-dimensional image from a 2-dimensional representation" with traditional conceptions of declarative memory (i.e., with the "memory" account of rhinal cortex function that is used to dismiss other findings of rhinal lesion-induced deficits)? This does not seem plausible. I would like to see either a different, more compelling reason for attributing the Lee et al. findings (both the 2005 and 2010 studies) to a "memory" deficit, or an alternative interpretation of the present results that can accommodate all of the data in a satisfying way.

Authors: The reviewers suggest that the short-term memory demands imposed by oddity tasks are equivalent to the sensory/perceptual demands of the dynamic visual world. In the Lee et al., 2005 study.… the sum of the saccadic intervals between the different objects, and among features within each object, will be on the order of 100s of ms, during which information has to be actively held in some form of short-term memory.

Reviewer: Again, it is not clear how the authors' argument rebuts the reviewer's concern. If rhinal cortex lesions impair the ability to hold information in memory for ~100ms, this would have serious deleterious effects on perception of the dynamic visual world. For example, when a prime stimulus disrupts perception of a subsequent target stimulus, its effects can either blend with (boost) or be "discounted" from (detract from) perception of the target stimulus, depending on for how long the prime appears. The prime duration at which our perceptual systems tend to switch from blending to discounting is approximately 100-300ms (e.g., Huber, 2008). In other words, basic mechanisms of dynamic perception would be massively altered if the ability to maintain information for 100ms were lost. This is not typically how the experience of individuals with rhinal cortex lesions is characterized.

Reviewer #2:

I am satisfied with the revision. The authors have responded adequately to reviewer comments.

Reviewer #3:

The authors have addressed the concerns raised in my review from the previous round, and the manuscript has been improved. I am satisfied with their revisions and response.

---

## [Author Response]

Essential revisions:Overall, this is an important study which is expected to be of interest to a broad array of visual, cognitive and behavioral neuroscientists. At the same time, however, there are substantial concerns about confounding factors in the task design and behavior that need to be discussed and addressed. In addition, improvements and additional analyses are needed for a more rigorous evaluation and a more complete picture of the issues under study.1) Task-design:1a) In the two-interval forced choice, cat/dog categorization task, is the first choice target always red and the second choice target always green? Or is the order of these two colored targets counterbalanced? In other words, is the first interval always correlated with the red target for cat stimuli and the second interval always correlated with green target for dog stimuli? Are monkeys using color to do the task? Or are they relying on an unknown combination of color and interval, so that it is not possible to know for sure what the monkeys are relying on since they are correlated?

Yes, the first choice target is always red, and the second always green. The first interval is always correlated with the red target, and the second interval with green. This structure does not influence the monkey’s decision, as the trial type (more cat-like or more dog-like) is selected on a random basis. To illustrate this point: Minamimoto et al., 2010, used a similar task structure to that used in the present study; when they substituted the categorical cues for stimuli derived by randomly arranging black and white pixels, the monkey’s performance fell to chance.

It is extremely unlikely that monkeys use the timing of the interval to perform the task. The timing of the color change from red to green varies from 2000 – 3000 ms during the testing phase (500 – 1500 ms during the initial training phase), and the timing window for correctly releasing the bar after the target changes to green is 200 – 1000 ms. These timings encourage the monkeys to use the color cue to determine when to make their responses; a strategy based on timing alone would result in very low levels of success. We explain this, with further discussion, in the first paragraph of the subsection “Behavior”.

The dependence of the reaction time distributions on the difficulty of the cat-like options indicates that the monkeys are releasing the bar when they have presumably gathered enough information to make a decision. The low variance across morph levels for bar release responses after the color change from red-to-green (Figure 3B, C, D) confirms that the monkeys are attending to the color change as opposed to using a timing strategy.

1b) Clarification in text: In the subsection “Experiment 1 – learning to categorize morphed stimuli”, where the task is described, it is stated that monkeys were rewarded for responding with a lever release for cat-like stimuli in the red (first) interval and for dog-like stimuli in the green (second) interval. However, in the second paragraph of the subsection “Behavior”, it states that only responses for dog-like stimuli during the green (second) interval were rewarded and there was never a reward for releasing while the red target was present. There is a discrepancy here and needs to be resolved.

Thank you for bringing this error to our attention. The description in the subsection “Behavior” was correct. We have modified the earlier description of the task to reflect this.

2) Psychophysical results (Figures 2, 5 and 6):If the task contains an asymmetrical reward structure in which monkeys are rewarded only for the "dog" category, can the results in Figure 2B be explained by this reward bias? It looks like psychophysical performance for TE lesioned monkeys is impaired only for 0-20% dog stimuli (which are essentially cat-like stimuli). It is possible that monkeys are very good at the category that is rewarded and not good for the category that is unrewarded. Bias in performance is also present only for TE lesioned monkeys and not for controls or rhinal lesioned monkeys. Further, in Figures 5 (5B and 5D) and 6 (6B), the patterns of deficits in TE monkeys is biased towards the left of the psychophysical curve (just like in Figure 2). More insight is needed to understand whether this bias is a result of the asymmetrical reward structure? Does the bias switch if only cat stimuli were rewarded?

The asymmetrical reward does produce a response bias. However, this bias is present for all groups, and hence cannot account for the deficit in the TE group. We have emphasized the universal effect of the bias in the first paragraph of the subsection “Experiment 2 – area TE removal impairs perceptually difficult categorization”.

We provide further insight into the reason for the direction of the bias (to the left of the psychophysical curve) in the subsection “Experiment 3 – area TE impairment with visually degraded stimuli”.

We did not test whether the bias switched if we switched reward contingency between the two categories, however we would strongly expect the bias to favor the rewarded category.

Also, for TE lesioned monkeys (Figure 2B), saturating performance (at 80-100% morph levels) decreases with repetition but not for controls or rhinal lesioned monkeys. Is this statistically significant?

The decrease in saturating performance (at 80-100% morph levels) with repetition for the TE group does not reach statistical significance (RM-ANOVA on the parameter of the curve fit that corresponds to saturating performance – ‘*b*’ in the function: *a* + *b* / (1 + exp(*c* * x + *d*)): main effect of treatment group, p = 0.99; main effect of session, p = 0.61, interaction of group*session, p = 0.51).

3) Behavior and training for Figure 2:What was the frequency of repetitions for the three groups of monkeys? Was it one repetition per day and was it similar across all the individual monkeys in the three groups? Could the impaired learning in TE lesioned monkeys be a result of more repetitions within the same time interval for control and rhinal lesioned monkeys but lesser repetitions within the same time interval for TE lesioned monkeys?

The frequency of repetitions was identical for all treatment groups. We describe this in greater detail in the last paragraph of the subsection “Behavior”. All monkeys received the same number of list presentations (i.e. number of trials), and there was no consistent difference between groups in how much time was required to complete the testing sessions.

4) Analysis for Figure 3:The text (subsection “Experiment 2 – area TE removal impairs perceptually difficult categorization”, first paragraph) mentions that the analysis used only sessions 11-20 in which asymptotic performance was reached to remove the effect of learning. However, in the subsection “Experiment 1 – learning to categorize morphed stimuli”, it states that while controls and rhinal lesioned monkeys reached asymptotic performance in 10 sessions, TE lesioned monkeys reached asymptotic performance in 14 sessions. Yet, to compare across groups for Figure 3, sessions 11-20 have been used for analysis, even though TE lesioned monkeys did not reach asymptotic performance by session 11. It might be cleaner to report analysis for Figure 3 using sessions in which all monkey groups reached asymptotic performance (perhaps session 14-20). The results in Figure 3A could be attributable to slower learning in TE.

In Experiment 1, the TE-removal group were slower to learn, only reaching the level of performance of controls by the 14^th^ repetition of the stimulus set; *all* sessions were included in the analysis. In Experiment 2 we wanted to exclude the potential confound of a learning effect by evaluating performance after a level plane had been acquired. In Experiment 2, all monkeys had reached asymptotic performance by the tenth session, hence sessions 11 – 20 were used in the analysis. We have clarified this in the first paragraph of the subsection “Experiment 2 – area TE removal impairs perceptually difficult categorization”.

5) Reaction-time distributions in Figure 3:In Figure 2C, the variance in RTs for release on red is lower than the variance in RTs for release on red in Figure 2B. Also, in Figure 2C, the variance in RTs for release on green is higher than the variance in RTs for release on green in Figure 2B and 2D. In both the lesioned monkey groups (2C and 2D), RTs for release on red are faster on the hardest conditions (near boundary) as compared to controls. Is there an explanation for these effects?

The plots of the individual monkey’s reaction times (now provided in Figure 3—figure supplement 2) go some way to explaining the differences in variance noted by the reviewers; for example, one of the TE-lesioned monkeys (monkey ‘G’) was almost 200 ms slower than all of the other monkeys in all treatment groups to release the bar in response to the appearance of the green target. However, the success rate of this monkey in classifying the morphed images correctly was between those of the other two TE-lesioned monkeys (see panel A of Figure 3—figure supplement 2), suggesting that the slower reaction time cannot easily be interpreted to explain the deficit.

Reaction times were not statistically different among groups. The reviewers noticed a trend towards slower reaction times for the control group at the most ambiguous levels of category membership, but the statistical analysis indicates that this was not significant.

6) Color vision related to Figure 4:Previous studies (Buckley 1997) have shown that TE lesions cause deficits in color discrimination. Do the anatomical locations of TE lesions in this study match with other studies that show color deficits? Does the TE lesioned group have difficulty in discriminating color? And does this affect their performance on the task? In the first paragraph of the Discussion, it mentions that these monkeys can detect the change from red to green. Is this from a different task? This should be explained in the text or referenced in some form.

The TE lesions in the present study are larger than those reported in Buckley, 1997, in that they extend medially to the lateral bank of the occipitotemporal sulcus. We have reproduced the figure illustrating TE lesion extent for simple reference (Figure 1—figure supplement 1). The reference to red-to-green color discrimination refers to the experiments described in the present study, we clarify this in the first paragraph of the Discussion. If the TE-lesioned group couldn’t discriminate color, they wouldn’t be able to complete the preliminary training phase in which monkeys are required to report the change of the target color from red to green.

TE-lesioned animals learned the red-green color discrimination at the same rate as controls and those with rhinal lesions. We did not test whether the monkeys were discriminating color or luminance. What is important is that their responses were governed by the test image.

7) How does the size of the TE lesion compare to the size of the rhinal lesion? For example, if TE lesions are larger than rhinal lesions, this could potentially lead to more deficits in TE lesioned groups than rhinal lesioned groups.

The TE lesions result in the removal of a larger volume of tissue than the rhinal removals. However, others have demonstrated that TE removals do not non-specifically produce greater impairments than rhinal cortex removals (Buckley et al., 1997, Buffalo et al., 1999). More importantly, the rhinal removals are equivalent to those previously reported – specifically in studies in which the authors reported a deficit in visual perceptual processing, such as the Bussey et al., 2003 study – which used exactly the same boundaries for rhinal cortex as were used in the present study. We have emphasized this in the second paragraph of the Discussion.

8) In both groups of lesioned monkeys, is there a reduction in attention or motivation during task performance? For instance, are more trials aborted before completion or are there more fixation breaks – and do these differ between monkey groups?

Motivation and attention do not appear to be affected by either lesion type. This statement and supporting evidence are presented in the last paragraph of the subsection “Experiment 2 – area TE removal impairs perceptually difficult categorization”. Monkeys were not head-posted, hence no fixation data is available. Sessions were limited to a maximum of 440 – 500 trials per day (depending on the experiment). In other studies in which we allow the monkeys to work to satiety, they will routinely work for 2000 – 3000 trials a day (depending on the individual), hence we had no difficulty keeping all three groups of monkeys motivated and attentive throughout testing.

9) Please include plots of the data from individual TE animals for all of the figures where a deficit is shown (in order to evaluate the heterogeneity of performance, and to see in which animals a reward bias could account for their behavior). Without seeing the TE lesions (including subcortical structures), we cannot know if they encroached upon reward circuitry or other brain regions to a greater extent than the Rh lesions, allowing non-perceptual explanations of the deficit to remain possible.

We have included plots of individual monkey performance for all experiments in which a deficit was reported (Figure 3—figure supplement 2; Figure 5—figure supplement 1; Figure 6—figure supplement 2). We have also reproduced the reconstructions of the TE lesions in Figure 1—figure supplement 1.

10) Please clarify how the present task differs from Task 1 of Lee et al., 2005, and how it has reduced memory demands relative to Lee and Rudebeck, 2010. There is a large prior literature showing Rh lesion-induced perceptual deficits that may be prematurely dismissed on the basis that those tasks did not eliminate memory demands adequately. But those tasks were described as having almost non-existent memory demands. Lee and Rudebeck, 2010, used a "possible/impossible" object judgement with no memory demands. Similarly, Lee et al., 2005 used a task almost identical to the present monkey task except that it displayed 2 stimuli simultaneously, with an identical category judgment (referring to a 'target' category defined at task outset); thus, in Lee et al., the greatest memory demand was comparing to a target category held in mind (as in the present study), not comparing between two adjacent stimuli. Second, it seems implausible that the extremely short memory demands imposed by prior oddity tasks (saccading between adjacent stimuli) are enough to render Rh involvement critical. This implies a role for Rh cortex in sensory/perceptual integration so extreme that patients with Rh damage should be unable to make sense of the dynamic visual world. Please provide further discussion of this important point. You may not be able to fully resolve this as it will likely remain hotly debated but describing your specific concerns in a convincing and detailed way will go much farther to increase the impact of this paper.

Both tasks performed by the subjects in the Lee at al. (2005) study are significantly different in design to those presented in the current study. Lee ‘task one’ is a classic visual discrimination task (simultaneous S+ vs. S-), where the categories offer no information that can help the subjects solve the task. The categories in Lee et al. are used to assess performance at different levels of perceptual difficulty, and to determine whether stimuli in different semantic groupings are processed in different brain regions. Lee ‘task 2’ is something of a hybrid between visual discrimination and oddity tasks; the subject can use one stimulus to determine which of the other two stimuli is most perceptually similar (i.e. the target), and thus must make real-time comparisons among the 3 stimuli presented (further discussion of this point below). ‘Task 2’ could also be solved using the same visual discrimination learning required in ‘task 1’, although we agree with the authors that it is parsimonious to assume that the adoption of the former strategy is more likely. In both tasks, the subjects are being asked to make a stimulus-reward association – task 1 over the longer timescale of multiple presentations; task 2 over the shorter timescale of within-trial presentations. Our task places no similar requirement for ‘stimulus-reward’ mapping on our subjects. The conclusion we draw from these comparisons is that Rh cortex is likely critical for stimulus-reward association memory (as others have demonstrated), but that memory for ‘category’ is supported elsewhere (earlier in the visual system). We feel that to elaborate this point in full in the Discussion of the present study would oblige us to engage in a similar depth of analysis for several of the other equally well-designed studies performed in human subjects, and hence would distract from the focus of the manuscript; we have included a reference to the Lee et al., 2005 paper in the section of the Discussion in which we acknowledge the debate surrounding the role of rhinal cortex in the human literature (Discussion, second paragraph).

The reviewers suggest that the short-term memory demands imposed by oddity tasks are equivalent to the sensory/perceptual demands of the dynamic visual world. In the Lee et al., 2005 study, and all other robust studies on this topic, the authors are careful to ensure that no single feature can be used to easily discriminate among the stimuli presented. That means that in an oddity task the subject must attend to a minimum of three images, and must attend to a minimum of two features in each of those three images (and probably more than two features to be confident). If this task requires foveal vision, as seems likely, then the sum of the saccadic intervals between the different objects, and among features within each object, will be on the order of 100s of ms, during which information has to be actively held in some form of short-term memory.

Lee and Rudebeck, 2010. implemented a task that required subjects to report whether drawings of stimuli were viable as 3D objects. The memory demands of that task are much more similar to those of the present study – the subjects classified stimuli into one of two categories. However, the task only tests ‘perception’ in an abstract sense; there remains a confound with cognitive load – the mental reconstruction of a 3-dimensional image from a 2-dimensional representation demands more than simple perception of the object as a whole.

We certainly agree with the reviewers’ comment that this topic will be hotly debated! We have modified the second paragraph of our Discussion to detail our argument (we hope) with greater clarity.

[Editors' note: further revisions were requested prior to acceptance, as described below.]

This revision was very responsive in addressing several questions posed by the reviewers and providing further clarification of the design and interpretation of the reported results. There was not complete agreement across the reviewers as reviewer 1 suggests an intriguing alternative interpretation for the results that we hope will be the basis for further discussion. Their re-review is included below and we would ask that you consider their comments and make some final revisions to your paper to respond to their concerns.

We would like to thank reviewer 1 for the detailed constructive criticism, thoughtful arguments, and helpful suggestions.

As a general statement, we would like to note that the key observation of interest in this manuscript is the *lack* of impairment in performing perceptually-difficult categorizations in monkeys with rhinal removals. In this context, the modest impairment of the group with TE removals serves as a positive control.

Reviewer #1:I remain unconvinced that the results and their interpretation are sufficiently compelling for publication in eLife (but I accepted the earlier, collective "letter to the authors" that was reasonably positive, because I see that I might be in the minority). My reservations stem from the ambiguous nature of the task (and the pattern of results in individual TE monkeys) and from weaknesses in the arguments for the key interpretations/conclusions. I respond to the authors' response letter below.Authors: We have included plots of individual monkey performance for all experiments in which a deficit was reported (Figure 3—figure supplement 2; Figure 5—figure supplement 1; Figure 6—figure supplement 2).Reviewer: This confirms what I ascertained by looking at the raw data: the TE lesion effect comes almost entirely from one monkey (K). In the other 2 TE monkeys, to the extent that they show any deficit, it is in classifying cats as dogs without a symmetrical tendency to classify dogs as cats. In other words, in these two monkeys, their small number of errors could be due to response bias (greater propensity to release the bar on green, because of increased reward-seeking/reduced risk-aversion, etc.). The authors state in the revised manuscript that "Due to the asymmetrical reward structure of this and similar tasks, when monkeys fail to discriminate between the two offers they will usually resort to releasing the lever during the presence of the green target 100% of the time, as this is the only condition in which reward is available." This illustrates the ambiguous nature of the task: if monkeys show a bias toward releasing on green (i.e. selecting "dog") it could be due to failure to discriminate or due to response bias (since the assignment of reward contingencies is not symmetric with respect to cat/dog category) and we cannot know which. This weakness is a key reason why I do not feel the present results are compelling enough to warrant publication in eLife.

We thank the reviewer for highlighting this issue for further clarification. If a response bias were introduced by a specific lesion (e.g. TE removal), that bias should be observed across all experimental conditions. The bias that exists (for all monkeys – see subsection “Experiment 2 – area TE removal impairs perceptually difficult categorization”, first paragraph) is only exacerbated in the TE removal group when the task is made perceptually more difficult in Experiment 3 (i.e. by partially obscuring the stimuli used in Experiment 2 with masks – Figure 5B). In this case, the bias (upward translation of the fitted function) is seen because of the increased perceptual difficulty. The increased perceptual difficulty is manifested as flattening of the slope of the discrimination function).

All three monkeys with TE removals (K, T and G) showed flattening of the discrimination curve relative to all control and all rhinal-lesioned monkeys in all of the morph discrimination tests. Critically, the gradients of the discrimination curves for the control monkeys and those with rhinal-removals were indistinguishable.

When the discriminations are made more difficult (Experiments 3 and 4) there are two changes in the discrimination functions. First, the TE monkeys’ discrimination performance becomes considerably worse as shown by the decrease in slope of the discrimination curve. Second, as correctly noticed by the referee, the monkeys have changed their criterion for classifying a stimulus as a dog as shown by the shift in the 50% discrimination point to the left. Since these changes occur when the discrimination is made difficult by the masks, we can conclude that the monkeys have a discrimination deficit, and in the face of the discrimination being made more difficult the monkeys have changed their criterion by classifying more trials as dogs. The change in criterion is not surprising in the face of the asymmetrical reward structure of the task.

Authors: Lee et al., 2001, 'task one' is a classic visual discrimination task (simultaneous S+ vs. S-), where the categories offer no information that can help the subjects solve the task.… Lee 'task 2' is something of a hybrid between visual discrimination and oddity tasks; the subject can use one stimulus to determine which of the other two stimuli is most perceptually similar (i.e. the target), and thus must make real-time comparisons among the 3 stimuli presented (further discussion of this point below). 'Task 2' could also be solved using the same visual discrimination learning required in 'task 1', although we agree with the authors that it is parsimonious to assume that the adoption of the former strategy is more likely. In both tasks, the subjects are being asked to make a stimulus-reward association – task 1 over the longer timescale of multiple presentations; task 2 over the shorter timescale of within-trial presentations. Our task places no similar requirement for 'stimulus-reward' mapping on our subjects. The conclusion we draw from these comparisons is that Rh cortex is likely critical for stimulus-reward association memory (as others have demonstrated), but that memory for 'category' is supported elsewhere (earlier in the visual system).Reviewer: The authors argue that the key distinction between the Lee et al. tasks and their task is the use of only a single item per learned discrimination (in Lee et al.) versus the use of sets of items constituting the to-be-discriminated categories (i.e. multiple cats and multiple dogs) in their task. In Lee et al., the requirement to associate each single item with reward thus renders the task a "memory" task. Whereas, in the authors' task, because monkeys had to generalize across multiple items within a category and correctly associate that category with reward, the task is not a memory task but a perceptual discrimination task. Note that "and correctly associate that with reward" was added by me – the authors make no mention of reward in describing their own task.

We agree with the reviewer that it is important that we clearly convey that our task *does* include a memory demand for category boundary/prototype. We currently state that our task “requires memory for a categorical exemplar or boundary, along with the category-response mapping”. However, the key distinction we wish to make is that our task carries a requirement for category memory, and not short-term memory, or item-by-item stimulus-reward associations (of the type required for standard S+/S- visual object discriminations). We now state this explicitly in the summary of experimental design: “All tasks required remembering visual perceptual categories. However, in every trial the monkeys responded while the stimulus was present, thereby minimizing demands on short-term memory”.

Yet an association between the category and the reward is critical for correct performance in the authors' task, and in fact this category-to-reward mapping is more complex than in the Lee et al. tasks (monkeys must learn: if "dog", then release while green; if "cat", then do not release until red). To argue that a simpler reward contingency (in Lee et al.) makes the task more "mnemonic" than a complex reward contingency (in the present task) seems backwards.

The reviewer is correct that the basic rules of the two tasks are different. However, the category-action mapping in our task (if ‘dog’ release while green, etc.) has been learned by all groups, as evidenced by their near 100% accuracy at the extremes of the morph scale (i.e. 0% dog and 100% dog in Figure 3A). A similar type of response mapping to one of two intervals is required to perform the contrast sensitivity task, and all groups performed this without bias. In any case, the learning of the rule is not the ‘mnemonic’ component we refer to here. We agree that our task and that of Lee et al. place different memory demands on the subject. Our task requires the monkeys to learn a category boundary, using a large set of fixed stimuli. In contrast, Lee et al. required humans to learn specific stimulus-reward associations, with two stimuli at a time, over a small number of repetitions. In fact, these differing memory demands form the crux of our argument: if both tasks are perceptually challenging, and differ significantly only in their mnemonic requirements, then the presence of a Rh impairment in their task, and absence in ours, seems most parsimoniously interpreted as due to the different mnemonic demands. This leads to the conclusion that Rh cortex is only required for certain types of memory (e.g. for single items and not for categories).

I agree that what makes the present task different from the Lee et al. tasks is the requirement for generalizing across category exemplars, but this implies a very different interpretation than the one offered by the authors. The most sensible interpretation is as follows. The present task is a categorization task that requires both generalization across diverse cats (or across diverse dogs) and discrimination of cats from dogs. Therefore, the optimal representations are not whole, unique objects (residing in rhinal cortex), but rather Shimon Ullmann-esque intermediate complexity features (known to reside in IT) that allow generalization across distinct cats as well as discrimination of cats from dogs.

This is indeed a thought-provoking alternative explanation of our results. We have added a section to the Discussion to cover this topic (Discussion, second paragraph).

This accords with another interesting/complicating factor in the present task: owing to the distribution of perceptual features possessed by cats and dogs, most cats are a plausible subset of dogs, but most dogs are not a plausible subset of cats (Mareschal, French and Quinn, 2000; Mareschal, Quinn and French, 2002). Given this category asymmetry, having compromised IT representations (needed for generalization and discrimination) might lead to a bias toward classing cats as dogs, as seen in the data. One way to test this would be to replicate the study in a design without reward asymmetry, to see if the dog-bias (which could then only arise from inherent category asymmetry) still exists.

It has been suggested that the observation that cats are a plausible subset of dogs but not vice versa is driven by the greater variability among dogs. The studies the reviewer references were performed on human infants, with very small sample sets (12 stimuli per category). The training sets used in the present study were much larger (960 stimuli per category [subsection “Behavior”, second paragraph]); thus, the corresponding reduction in variability makes the ‘plausible subset’ interpretation unlikely. With regards to the potential effects on discriminability associated with bias, please refer back to our response to reviewer 1.

Authors: Lee and Rudebeck, 2010, implemented a task that required subjects to report whether drawings of stimuli were viable as 3D objects. The memory demands of that task are much more similar to those of the present study – the subjects classified stimuli into one of two categories. However, the task only tests 'perception' in an abstract sense; there remains a confound with cognitive load – the mental reconstruction of a 3-dimensional image from a 2-dimensional representation demands more than simple perception of the object as a whole.Reviewer: Here, it is not clear how the authors' argument rebuts the reviewer's concern. Do the authors mean to equate "mental reconstruction of a 3-dimensional image from a 2-dimensional representation" with traditional conceptions of declarative memory (i.e., with the "memory" account of rhinal cortex function that is used to dismiss other findings of rhinal lesion-induced deficits)? This does not seem plausible. I would like to see either a different, more compelling reason for attributing the Lee et al. findings (both the 2005 and 2010 studies) to a "memory" deficit, or an alternative interpretation of the present results that can accommodate all of the data in a satisfying way.

We do not wish to equate ‘mental reconstruction of a 3D image’ with memory. Neither task places explicit demands on short-term memory. We merely note that this type of reconstruction demands more than simple perception of the object as a whole.

Authors: The reviewers suggest that the short-term memory demands imposed by oddity tasks are equivalent to the sensory/perceptual demands of the dynamic visual world. In the Lee et al., 2005 study.… the sum of the saccadic intervals between the different objects, and among features within each object, will be on the order of 100s of ms, during which information has to be actively held in some form of short-term memory.Reviewer: Again, it is not clear how the authors' argument rebuts the reviewer's concern. If rhinal cortex lesions impair the ability to hold information in memory for ~100ms, this would have serious deleterious effects on perception of the dynamic visual world. For example, when a prime stimulus disrupts perception of a subsequent target stimulus, its effects can either blend with (boost) or be "discounted" from (detract from) perception of the target stimulus, depending on for how long the prime appears. The prime duration at which our perceptual systems tend to switch from blending to discounting is approximately 100-300ms (e.g., Huber, 2008). In other words, basic mechanisms of dynamic perception would be massively altered if the ability to maintain information for 100ms were lost. This is not typically how the experience of individuals with rhinal cortex lesions is characterized.

The critical distinction we make between the short-term memory demands placed on subjects by the carefully controlled behavioral tasks claimed to test monkey perception vs. the types of ‘short-term memory’ required for seamlessly processing visual information in a dynamic visual world, is that the former is an active, effortful process, while the latter is a passive, and seemingly effortless one. As such, they will likely have very different neural substrates (see Wittig et al., 2016, and Wittig and Richmond, 2014 for a discussion about selective working memory vs. passive recency/novelty memory and their roles in monkey short term memory).